# The Mechanical Properties of Plant Fiber-Reinforced Geopolymers: A Review

**DOI:** 10.3390/polym14194134

**Published:** 2022-10-02

**Authors:** Chun Lv, Jie Liu, Guoliang Guo, Yanming Zhang

**Affiliations:** 1College of Architecture and Civil Engineering, Qiqihar University, Qiqihar 161006, China; 2College of Light-Industry and Textile Engineering, Qiqihar University, Qiqihar 161006, China; 3Engineering Research Center for Hemp and Product in Cold Region of Ministry of Education, Qiqihar 161006, China

**Keywords:** geopolymer, plant fiber, mechanical property, compressive, flexural, tensile

## Abstract

Both geopolymer and plant fiber (PF) meet the requirements of sustainable development. Geopolymers have the advantages of simple preparation process, conservation and environmental protection, high early strength, wide source of raw materials, and low cost. They have broad application prospects and are considered as the most potential cementitious materials to replace cement. However, due to the ceramic-like shape and brittleness of geopolymers, their flexural strength and tensile strength are poor, and they are sensitive to microcracks. In order to solve the brittleness problem of geopolymers, the toughness of composites can be improved by adding fibers. Adding fibers to geopolymers can limit the growth of cracks and enhance the ductility, toughness and tensile strength of geopolymers. PF is a good natural polymer material, with the advantages of low density, high aspect ratio. It is not only cheap, easy to obtain, abundant sources, but also can be repeatedly processed and biodegradable. PF has high strength and low hardness, which can improve the toughness of composites. Nowadays, the research and engineering application of plant fiber-reinforced geopolymers (PFRGs) are more and more extensive. In this paper, the recent studies on mechanical properties of PFRGs were reviewed. The characteristics of plant fibers and the composition, structure and properties of geopolymers were reviewed. The compatibility of geopolymer material and plant fiber and the degradation of fiber in the substrate were analyzed. From the perspective of the effect of plant fibers on the compression, tensile and bending properties of geopolymer, the reinforcing mechanism of plant fibers on geopolymer was analyzed. Meanwhile, the effect of PF pretreatment on the mechanical properties of the PFRGs was analyzed. Through the comprehensive analysis of PFFRGs, the limitations and recommendations of PFFRG are put forward.

## 1. Introduction

Nowadays, Fiber-reinforced cement-based composites are increasingly widely used [1]. However, the preparation of cement-based materials requires a large amount of energy, which does not meet the requirements of energy conservation and environmental protection [2,3]. Geopolymer has many advantages and broad application prospects, and is considered as the most potential cementitious material to replace cement [4,5,6]. The production process of geopolymers is relatively simple compared with traditional cement, which has outstanding advantages in energy saving and carbon emission reduction. Compared with organic polymer materials, geopolymers have the advantages of high hardness, high strength, good thermal stability, and strong antioxidant capacity [7,8].

In general, the synthesis of geopolymers requires two materials: active solid silica-aluminate precursor and initiator solution. The source of precursor raw materials is very rich. For example, kaolin, silica fume, fly ash, slag, and other industrial wastes [9,10]. The most commonly used is kaolin, fly ash, and slag. Geopolymers can be divided into industrial waste geopolymers and non-industrial waste geopolymers. The former refers to geopolymers synthesized from industrial wastes, such as fly ash and slag, while the latter refers to geopolymers synthesized from non-industrial wastes, such as kaolin [11,12]. The initiator solution acts as a binder, alkali activator and dispersant. The initiators include alkaline substances of sulfate, silicate, alkali metal hydroxide, and ammonium.

The mechanism of alkali excitation was put forward by Davidovits. According to Davidovits J, the reaction of geomeric is the reaction of Al-O and Si-O bond breaking and recombination under the catalytic action of alkali initiator. He stimulated geological minerals with alkali metal silicate solution under strong alkaline conditions to form polymeric aluminum silicate materials [13]. Subsequently, other solid silicate feedstocks, such as fly ash and pulverized blast furnace slag, were successfully prepared as geopolymers.

However, similar to most inorganic materials, geopolymers are very prone to cracking and carbonization despite the aforementioned advantages [14,15,16,17]. The flexural strength and tensile strength of geopolymers are poor due to the ceramic-like shape and brittleness of geopolymers. In addition, geopolymers are sensitive to microcracks. In order to solve the brittleness problem of geopolymers, the toughness of composites can be improved by adding fibers. Fiber-reinforced geopolymers (FRGs) can limit the growth of cracks and enhance the ductility, toughness, and tensile strength of geopolymers [18,19].

At present, the fibers used in composites mainly include metal fibers, inorganic fibers, synthetic fibers, natural fibers, etc. [20,21,22,23,24,25]. Synthetic fiber has better mechanical properties and its production cost is similar to that of steel fiber. In addition, the production process of synthetic fiber is easy to produce environmental pollution, and the degradation cycle after the end of the use cycle is long, which is difficult to meet the requirements of sustainable development. In the traditional fiber, the preparation of steel fiber needs a lot of resources and energy. Compared to carbon fiber, basalt fiber production cost is higher.

In contrast, PF is a good natural polymer material with advantages, such as low density and high aspect ratio [26]. It is not only cheap, easy to obtain, and abundant, but it can be repeatedly processed, and it is biodegradable. PF has the advantages of high strength and low hardness; adding it to composite can improve the brittleness of the material and also enhance the strength of the material. Therefore, PFs can be used as fillers to add to geopolymers [27,28].

In recent years, PF has been gradually used in the development and preparation of engineering materials to improve the brittleness and other properties of cementitious materials. Bast fibers are used as reinforcing materials in many PFs. Othuman studied the effects of the addition of kenaf, ramie, hemp, and jute fibers on the properties of lightweight foamed concrete. The weight fraction of PF was maintained at 0.45%. The results show that PF plays an important role in improving the durability of composites. In terms of workability, ramie caused the slump of the composite to decrease. In terms of porosity and water absorption, the addition of jute fiber had the best effect [29,30]. Abbas et al. [31] discussed the improvement of tensile strength and flexural strength of cementitious composites by appropriate content and length of kenaf fibers. Abirami et al. [32] added 0.25-1% of different types of kenaf or sisal fibers to cement-based composites. The results showed that the compressive strength and tensile strength of the two fibers increased by 6.5%, and 12.7%, respectively. The mechanical properties of 1% sisal or kenaf fibers reinforced composites are better than those of other fibers added composites. In addition, compared with sisal fiber reinforced composite, kenaf fiber reinforced geopolymer has higher compressive strength, but slightly lower flexural strength and tensile strength. Beddu et al. [33] found that kenaf fiber composites had higher compressive strength than composites containing polypropylene fibers. The tensile strength of polypropylene fiber reinforced cementitious composites increased significantly. Petrella et al. [34] used wheat straw and perlite beads of different lengths and contents as aggregates to study the performance of straw and perlite composite mortar, and compared the performance with that of conventional mortar. The workability of straw composites decreased with the decrease of fiber length and the increase of straw volume. It was found that the mechanical properties of straw composites increased with the increase of fiber length and decreased with the increase of fibers content. Ahmad et al. [35] believed that coconut fibers could improve the crack resistance of concrete, while the incorporation of synthetic fibers reduced the fluidity of concrete.

The difference with cement-based materials is that the reaction system of geopolymers is different from that of cement. The geopolymers contain only trace amounts of calcium hydroxide, which erodes PFs more violently than sodium hydroxide solution and is easy to mineralize fibers [36,37,38]. PFRGs can overcome the disadvantage of poor durability of composites, and its performance may be better than cement-based composites. The results show that PFs have the feasibility to replace synthetic fibers in geopolymers. In this paper, SCIE database, MDPI database and Baidu Academic database were used to search the recent articles about the compatibility of plant fibers and geopolymers and the properties of PFRGs. No reviews related to PFRGs were found, and the content was novel. This paper can make the industry personnel more comprehensive understanding of PFRGs research status and process so as to carry out more in-depth research and engineering application. The characteristics of PFs and geopolymer are briefly summarized, and the compressive, bending, and tensile properties of PFRGs are emphatically analyzed. The compatibility between fibers and matrix and the mitigation of fiber degradation in matrix are discussed. Finally, the effect of PF pretreatment on the mechanical properties of the composites was analyzed, and the improvement of the mechanical properties of PFRGs was summarized.

## 2. Properties of PFs

PF is one of the most abundant natural resources. PF has the characteristics of low cost, light weight, rough surface, strong adhesion, and biodegradable. PFs, such as rice straw, rice husk, crop straw, bagasse, shavings, wood shavings, bamboo shavings, are waste raw materials that are very common. PFs can be generally divided into bast fibers, leaf fibers, seed fibers, grass fibers, wood fibers, straw fibers according to its source and parts. Some PFs can form more than one type of fibers. For example, flax and kenaf have bast and core fibers, while coconut and oil palm have fruit and stem fibers [39]. The classification of commonly used PFs is shown in Figure 1 [40,41,42].

As a member of PFs, straw waste quantity is large, the source is wide, has the typical representative. In general, the chemical composition of different types of PFs varies greatly [43,44]. As a biological resource, PF is mainly composed of cellulose (C_6_H_10_O_5_)n, hemicellulose (C_5_H_8_O_4_)m, lignin C_9_H_10_O_3_(OCH_3_)_0.9~1.7x_, ash, pectin, tannins, pigments, and esters. Among them, cellulose, hemicellulose and lignin constitute the supporting skeleton of straw, which exists in the form of cellulose-hemicellulose-lignin bond in plant [45,46,47]. The lignocellulosic composition of PFs is shown in Figure 2. Cellulose is composed of microfibers, which form the reticular skeleton of the fiber cell wall. Hemicellulose and lignin fill between the fibers and microfibers, acting as adhesives and fillers. The properties of different PFs are shown in Table 1.

As can be seen in Table 1, PFs have good tensile strength, tensile modulus, and elongation at break. The density of PF is slightly greater than 1.0 g/cm^3^, and the volume content and mass content of PFs are slightly different when used as reinforcing fibers.

## 3. Geopolymers and PFRGs

Geopolymer is a kind of inorganic polymer material with a three-dimensional network structure composed of Si-O_4_ and Al-O_4_ tetrahedral units. Geopolymer is a kind of environmental cementitious material with low energy consumption and less pollutant emission in the production process.

### 3.1. Polymerization Mechanism of Geopolymer

Davidovits [13] used metakaolin as raw material and NaOH or KOH as alkaline activator to analyze the formation process of geopolymers. Firstly, under the action of NaOH or KOH strong alkali solution, the Si-O and Al-O bond of metakaolin precursor are broken, and a series of oligomers similar to polymer monomers are formed. These oligomers are Si-O_4_ and Al-O_4_. The oligomers dehydrate gradually under the action of strong alkali, and then polymerize, forming geopolymer with three-dimensional network structure. With slag, fly ash, and metakaolin active materials as precursors and NaOH or KOH as alkali activators, the reaction mechanism of the obtained geopolymers is expressed as follows; this is shown in Figure 3.

Duxson et al. believe that the polymerization of geopolymers can be divided into four processes: solution, diffusion, polymerization, and curing [71]. The solution process is the aluminum silicate raw materials dissolved in alkali initiator solution and produce a large number of silicon, aluminum monomer process. The diffusion process is the gradual diffusion of silicon and aluminum monomer from the surface to the inside. Polymerization is a process in which the dissolved silicon and aluminum monomers and the silicon monomers in the initiator rapidly undergo condensation reaction and form silica-aluminum oligomers. The curing process is that the gel phase gradually becomes geopolymer after solidification and hardening.

### 3.2. Compatibility of PFs with Geopolymers

The compatibility of PFs and geopolymers affects the properties of composites. The common contact surface produced by the contact between different materials is the interface [72]. The interface of PFRGs not only includes the geometric surface where the geopolymer matrix and PFs contact each other, but also includes the transition region of this geometric surface, which is a complex microstructure [73]. By adjusting the characteristics of the interface layer and the bonding state, the interface control of the composite can be realized, so that the composite can achieve the best performance. The interface adhesion performance between fiber reinforcement and geopolymer matrix is the most critical factor in interface control technology of composites [74,75]. Alomayri et al. [76] used layup technology to prepare cotton fabric reinforced geopolymers with different layers. The bending strength, bending modulus, impact strength and fracture toughness of 3.6, 4.5, 6.2, and 8.3 wt% cotton fibers reinforced geopolymers were investigated. The results show that adjusting the compatibility of fibers and matrix can effectively improve the mechanical properties of composites. Camargo et al. [77] evaluated the compatibility between geopolymer and PFs by referring to the evaluation method of compatibility between cement and PFs. Tan et al. [78] studied the compatibility between metakaolin-based geopolymers and 15 kinds of forest residual PFs. The results show that the geopolymers have good compatibility with wood fibers and non-wood fibers, and the compatibility of wood fibers is better than that of non-wood fibers. Figure 4 shows the polymerization temperature of different PFs as a function of time. The maximum polymerization temperature of fresh PFRG was lower than that of pure geopolymer, and the maximum polymerization temperature of fresh PFRG was delayed than that of pure geopolymer, indicating that the effect of adding PFs in geopolymer was inhibited. In addition, after the occurrence of the highest geopolymerization temperature, the geopolymerization curve of PFRG decreased more gently than that of pure geopolymer, which was also due to the inhibition of PFs.

### 3.3. Degradation Behavior of PFs in Geopolymers

Although PFs have good compatibility with geopolymer, they also have degradation behavior due to the influence of alkaline activator in geopolymer.

The amorphous components in PFs will be degraded in different degrees when exposed to alkaline environment. Wei et al. [79,80] divided the alkaline degradation of PFs in cement matrix into four steps: degradation of lignin and partial hemicellulose; the complete degradation of hemicellulose leading to the loss of integrity and stability of the fiber cell wall; removal of cellulose microfibrils; and failure of cellulose microfibrils that leads to complete degradation of PFs, as shown in Figure 5. Alkaline hydrolysis will lead to the decomposition of hemicellulose and amorphous regions of cellulose fiber chains and affect the loss of integrity of the fiber–matrix interface zone, thus affecting the mechanical properties of composites [81,82].

Rocha et al. [83] found that the use of PFs increased the compressive strength and stiffness of the binder within 7 days. After 28 days, the mechanical properties of the geopolymer paste decreased due to the degradation of PFs. Ye et al. [84] found that the alkaline degradation of hemicellulose significantly reduced the degree of geopolymerization. Compared with cement-based materials, the geopolymer matrix showed good adhesion to cellulose fibers without significant degradation.

## 4. Mechanical Properties of PFRGs

As mentioned earlier, PF has a long history. Studies have shown that PF has the advantages of reducing matrix shrinkage, shortening matrix curing time, and improving ductility and tensile strength of composites [85,86]. The strengthening effect of PFs on geopolymer is mainly manifested in the enhancement of toughness, while the compressive strength is generally reduced [87,88]. The properties related to the toughness of the matrix including flexural and tensile properties are improved obviously. The preparation process of PFRGs should comply with relevant technical standards. Taye et al. [89] prepared PFRGs slurry by stirring fly ash, red mud and alkali solution at a speed of 700 RPM for 30 min. Then, gradually added the minced hemp fibers with the length of 10–13 mm, and mixed until the slurry was uniform. The geopolymerization temperature of PFRGs slurry is low and temperature has little effect on fiber properties [78].

### 4.1. Compressive Strength

Compressive strength is an important index to measure the mechanical properties of building materials. The compressive strength of PFRGs mainly depends on the compressive strength of matrix materials, and is also related to the bond between matrix and PFs. The influence factors of PFs on geopolymer compressive properties include fibers content and fiber type. PFs are inexpensive, biodegradable, and readily available, and have great appeal as reinforcement in the field of geopolymers.

The content of PFs will adversely affect the compressive strength of geopolymers. Fonseca et al. [90] analyzed the feasibility of pine fibers, palm fibers, razorgrass fibers and jute fibers as geopolymer reinforcement materials. The compressive strength of the specimens with mass added 1.5%, 3.0%, and 4.5% was tested. It was found that the higher the fibers content, the lower the geopolymers compressive strength.

An appropriate amount of PFs content distributed evenly in the matrix can enhance the compactness of the composite, reduce the porosity and crack, and thus improve the compressive strength. Ayeni et al. [91] studied geopolymer composites of 0.5, 1.0, 1.5, and 2.0% different weight of coir fibers contents. The compressive strength was increased from 16.91 N/mm^2^ to 21.25 N/mm^2^ by adding coir fibers with 0.5% fibers content. A similar trend was also found in the study of Wongsa et al. [26,92]. Wongsa et al. reported that 0.5% coir fibers content increased geopolymer compressive strength. The addition of 1% coir fibers reduced the compressive strength of the material. The results showed that the dispersion of coir fibers in geopolymer matrix was poor under 1% fibers addition. However, Korniejenko et al. [93] found that the addition of 1% coir fibers to geopolymer composites increased the compressive strength. When coir fibers content is 1.5%, the strength of geopolymer composite is 16.94 N/mm^2^. Its compressive strength is similar to that of geopolymers without fibers. The presence of coir fibers has no significant effect on the compressive strength of the PFRG. Notably, the strength decreased when 2% coir fibers were added, indicating that the composite structure was less dense. The higher the fibers content, the lower the density of the sample, the more difficult the geopolymer matrix is to accumulate, and the higher the porosity, resulting in reduced compressive strength. Lazorenko et al. [94] randomly reinforced geopolymer composites with 0.25–1.0 wt% short-cut flax fibers. The addition of short cut flax fibers will lead to the reduction of compressive strength of the PFRG. Zhou et al. [95] found that the addition of cotton stalk fibers reduced the density and compressive strength of PFRGs. The addition of cotton stalk powder can effectively improve the compressive strength of PFRG. The effect of cotton stalk powder in PFRG is mainly filled and cemented. Workiye et al. [96] prepared 0, 0.1, 0.2, 0.6 and 1 wt% corn straw reinforced geopolymer composites and carried out compressive strength tests. The measured compressive strength ranges from 16 MPa to 27 MPa. The results indicated that the compressive strength of calcined kaolinite-based geopolymer could be improved by appropriately adding corn straw fibers.

As mentioned above, the different laying forms of fibers in the matrix directly affect the compatibility between fibers and matrix. It also affects the compressive strength of the composite. Assaedi et al. prepared different layers of geopolymers reinforced with 2.4, 3.0 and 4.1 wt% woven flax fibers by manual layup technique and tested their mechanical properties, such as flexural strength, flexural modulus, compressive strength, hardness and fracture toughness [97]. The increase of flax fibers content can improve the mechanical properties of flax PFRGs, and its mechanical properties are better than those of pure geopolymer. The relationship between fibers content and compressive strength is shown in Figure 6. It can be seen from the figure that [97], significantly different from other studies, the fibers content has a positive relationship with the compressive strength because the horizontally laid fiber fabric can directly absorb and evenly distribute the load on the cross section. It can be seen that when the content of flax fibers is 4.1%, the compressive strength of geopolymer increases from 19.4 MPa to 91 MPa. This significant increase in compressive strength is due to the fact that the interface between the linen fabric and the geopolymer matrix is not exposed to any shear loads, reducing the possibility of the linen fabric separating or delamination from the geopolymer matrix at high loads. In addition, the long fibers in the matrix can transfer the stress to another fiber, and the compressive strength of the long fibers (30 mm) are also higher than that of the short fibers (10 mm and 20 mm), but there are not many studies in this area [28].

There are many types of PFs. Similar to fibers content, different types of fiber have different effects on the strength of PFRGs [98]. Korniejenko et al. [93] added 1% of different types of PFs to the geopolymer mixture. The results showed that the compressive strength of the PFRG containing coir, cotton and sisal fibers was 26.55%. It was 14.68% and 1.53% higher than pure geopolymer, respectively. However, the compressive strength of the PFRG with lafite fibers was 44.87% lower than that of the pure geopolymer. This reduction is due to poor compatibility and lack of cohesiveness between laffia fibers and geopolymers. Similarly, Ribeiro et al. [99] found that the compressive strength of the PFRG added to bamboo fibers decreased by 50% due to the increase in porosity. In conclusion, in all studies related to natural short fiber reinforced geopolymers, when the PFs content exceeds 1% by volume or mass, the compressive strength of the PFRG decreases [100,101,102,103].

**Figure 6 polymers-14-04134-f006:**
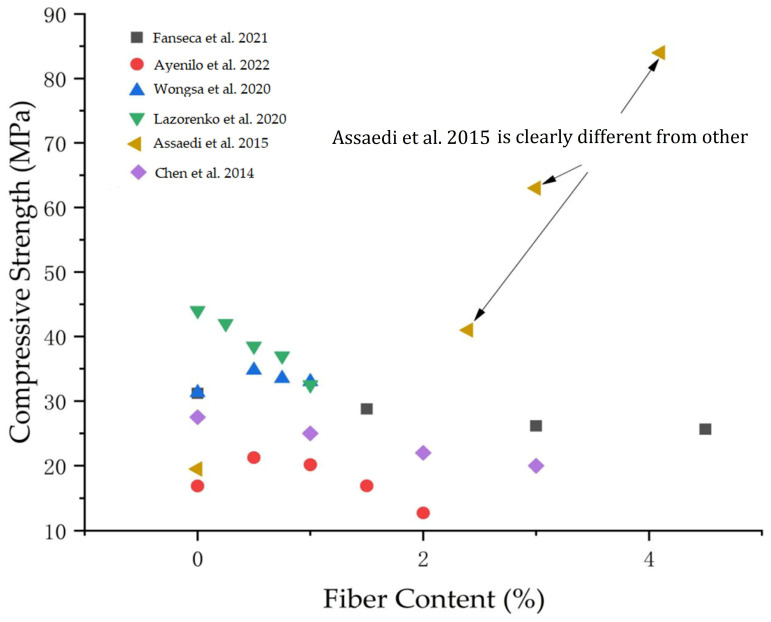
Relationship between fibers content and compressive strength of composites [87,90,91,92,94,97,101].

In the study of the compressive strength of fiber-reinforced composites, it is found that fiber-reinforced cementitious materials are similar to geopolymers. The increase of porosity caused by the addition of fibers to the matrix is the reason for the decrease of compressive strength. Ujiyama et al. found that adding sisal fibers to mortar had no significant effect on compressive strength. When the fibers content is 1.0%, the mechanical properties of concrete are improved in the initial and later stages [104]. In another study, the addition of 6.0% sisal fibers of length 40 mm to ordinary concrete resulted in a 22.0% reduction in compressive strength [105]. Similarly, kenaf fiber-reinforced concrete has better mechanical properties than standard concrete samples. When fiber intake increased to more than 1.25%, compressive strength decreased significantly [106]. However, Naraganti et al. observed that adding 1.5% sisal fiber to ordinary concrete increased its compressive strength by approximately 5.6% [107].

### 4.2. Flexural Strength

In terms of the properties of PFRGs, fiber type and fiber properties are more effective than matrix properties in improving the fracture properties, such as flexural strength, and fiber properties control the failure mechanism and fracture properties of composites. Ye et al. [84] found that 5 wt% lignin, cellulose and hemicellulose improved the flexural strength of geopolymers. The increase of lignin and hemicellulose content leads to the porosity, low density and brittle fracture of PFRGs, and reduces the flexural and compressive strength of PFRGs.

Because of the excellent properties of flax fibers, there are many studies on the use of flax PFRGs. The research results of Lazorenko et al. [94] showed that the flexural strength of the geopolymer increased from 4.0 MPa to 4.9 Mpa when the flax fibers were added to the geopolymer with 1.0 wt%. Different from brittle geopolymer substrates, the composites exhibit plastic failure characteristics and have high residual bearing capacity, and remain intact without failure even after being subjected to the maximum mechanical impact.

Among the PFs, cotton fibers have the highest content of cellulose. Alomayri et al. found that when 0.5 wt% cotton fibers were added to the geopolymer, the flexural strength increased from 10.4 Mpa to 11.7 Mpa. The flexural strength and fracture toughness are improved. However, the addition of more cotton fibers (0.7 and 1.0 wt%) resulted in a decrease in flexural strength [108]. Zhou et al. [95] found that the flexural strength of the geopolymer added with cotton stalk fibers were slightly increased. With the increase of fibers content, the compressive strength and flexural strength of the PFRGs increased first and then decreased, reaching the maximum value when the fibers content was 0.7%. It can be seen that when the fibers content continues to increase, with the increase of the fibers content, the density of the PFRGs decreases, the porosity increases, and the fibers tend to aggregate.

Wongsa et al. [92] added sisal fibers and coir fibers in different proportions to the geopolymer, and the volume fractions were 0, 0.5, 0.75, and 1.0 wt%, respectively. The mechanical properties of PFRGs were tested and compared with synthetic fibers and control pure geopolymer mortar. The results showed that the tensile and flexural strength properties of sisal fibers and coconut fibers were significantly improved by adding sisal fibers and coir fibers as reinforcement materials, similar to the use of glass fibers. At the same time, workability and compressive strength values have a downward trend. The addition of fiber increases the flexural strength and splitting strength of geopolymer mortar. This is due to the high tensile strength and elastic modulus of the fibers, and also because the stress in the sample can be transferred to the fibers through the interface with the geopolymer matrix [108,109]. The experimental results also show that the flexural strength and splitting strength are improved with the increase of fibers content, but the limit is when the fibers content is 1.00%. The workability of the mixture with volume fraction more than 1.00% is low, and it is difficult to cast and densification. The flexural strength of sisal fibers and coir fibers reinforced geopolymer mortars ranged from 5.3 to 6.6 Mpa, which was higher than that of pure geopolymer (3.1 Mpa) and synthetic fiber reinforced geopolymer (3.1 to 3.7 Mpa).

Figure 7 shows the relationship between flexural strength and fibers content. It shows that the fibers contents is positively correlated with the flexural strength in a certain range. However, when the contents of different fibers exceeds a certain value, its strength decreases obviously.

In general, fiber resistance to crack deflection, debonding, and bridging cracks slows down crack growth and increases fracture energy in fiber-reinforced composites [97]. In Figure 8, SEM images show the imprints of coir fibers (a) and bamboo fibers (b) on the matrix, indicating good contact between matrix and fibers. The rough straight line feature on the matrix can also be seen on the image, which indicates the mechanical interlocking and friction between the fibers and the matrix when the fiber is pulled out. Figure 8a shows that fiber pull-out is a toughening mechanism. In contrast to pure geopolymer, cracks in the geopolymer matrix deflect around the fibers rather than tear it, thus toughening the composite. The crack deflection mode at the fiber–matrix interface has the characteristics of a weak interface.

Different types of PFs also affect the flexural properties of the PFRGs. Ayeni et al. [91] added different proportions of coir fibers to improve the flexural strength of PFRGs. When the fibers content is 0.5%, the flexural strength is 9.74 N/mm^2^. When the fibers content is 1%, the flexural strength can reach 10.39 N/mm^2^. Due to the high tensile strength and elastic modulus of the coir fibers, stress may be transferred from the sample to the coir fibers through the interface of the geopolymer matrix. When the PFRG is subjected to bending load, the bending moment causes tensile stress. At the fiber–matrix interface, the tensile stress is converted to shear stress, resulting in resistance at the fiber–matrix interface. With the increase of fibers content, microcracks will form inside the composite structure. In contrast, another flexural strength test showed that the performance of the specimens without fiber was basically the same as that of the specimens with coir fibers, cotton fibers, or sisal fibers. The addition of natural fibers has no significant effect on the flexural strength of the PFRG, contrary to the addition of man-made fibers [98].

Kavipriya et al. [111] used 10, 20, and 30% bamboo sticks in place of coarse aggregate. At the same time, different proportions of sisal fibers were added, and the volume ratios were 0.25%, 0.5%, 0.75%, and 1.0%, respectively. The results showed that sisal fibers could effectively improve the flexural strength of PFRGs, as shown in Figure 9.

### 4.3. Tensile Strength

Tensile strength is a fundamental and important property of geopolymer composites. Because of their brittleness, geopolymers are very weak in tension. When the geopolymer is subjected to tensile load, cracks form rapidly [112]. The tensile properties of geopolymer can be improved by adding PFs.

Gianmarco et al. [113] compared the mechanical properties of PFRGs with different fibers contents (1, 2, and 3 wt%). The PFRG has high strength and fracture resilience, and the effect is maximum when the fibers content is 2 wt%. Mourak [114] extracted cellulose fibers from jujube stems and coated metakaolin particles formed by amorphous cellulose II-NA greatly reduced zeolite formation. The presence of cellulose II-NA decreases the porosity and increases the mechanical strength and density.

The flexural, compressive, and splitting tensile strengths of PFRGs under different flax tows contents are shown in Figure 10. The flexural strength of the PFRG increases because the tensile stress is transferred from the matrix to the fibers. Fibers in PFRGs prevent crack formation by dispersing loads. At the late stage of PFRG deformation, the fibers begin to be pulled out or deformed. Obviously, the destruction of PFRGs increases with the increase of fibers content. However, according to Figure 10a, a low content of fibers (0.25 wt%) does not significantly reduce the strength of the PFRG under static bending stress. It is smaller than the unreinforced matrix. Compared with pure geopolymers, the PFRG has compressive properties and the splitting tensile strength also shows a decreasing trend. These measurements did not show a clear pattern dependent on fibers content. See Figure 10c. This may be due to the randomness or orientation of the fibers. When 1.0 wt% fibers was added, the splitting tensile strength approached that of the unreinforced pure geopolymer, and there was a local decrease in the splitting tensile strength of the PFRG at a fibers content of 0.75%. Figure 10c can be explained by the sharp increase in hole size and the decrease in blending uniformity occurring at exactly the content (0.5–0.75 wt%) interval [94]. As PFs content increased to 1.0 wt%, the total tensile and tensile resistance became quite large, which compensated for the presence of these defects reflected in the increase of tensile strength at splitting.

### 4.4. Load-Displacement Behavior and Toughness

Geopolymers have attracted a lot of research interest as sustainable materials. However, as ordinary Portland cement, geopolymers exhibit a brittle behavior with low tensile strength, ductility, and fracture toughness.

The mechanism of improving toughness includes crack arrest at the fiber–matrix interface and increasing crack path through fibers with high aspect ratio [115]. Normally, when a crack starts to appear, all the tension at that location is carried by the fibers. If there is an increase in the force the fibers can withstand without breaking or pulling out, a new crack will appear at a different location. As a result, the fibers in the area will be activated and the force will be transferred. This process results in multiple cracking until the fiber fails or is pulled out of the matrix.

Chen et al. [101] found that the unit weight of sweet sorghum PFRGs decreased with the increase of fiber content. Although the addition of sweet sorghum fibers reduced the unconfined compressive strength slightly, the splitting tensile strength, bending strength, and post-peak toughness increased first and then decreased with the fibers content reaching 2%. The splitting tensile test also clearly showed that the brittle failure of the pure geopolymer changed to the ductile failure of the sweet sorghum PFRG. Figure 11 shows typical load and displacement curves of PFRGs with different fibers content in sweet sorghum during splitting tensile test. The pure geopolymer fails suddenly at peak load, while the addition of fibers significantly improves the post-peak ductility. With the increase of sweet sorghum fibers content, the peak load first increased and then decreased.

This is consistent with the previous argument that the main function of fibers is not to improve the compressive strength of PFRGs but to improve their toughness and control the further development of cracks in the matrix.

Alomayri et al. [108] found that the fracture toughness of PFRG reinforced with 0.5 wt% cotton fibers was 1.12 Mpa higher than that of pure geopolymer. This remarkable enhancement of fracture toughness is due to fiber pull-out, fiber breakage, and fiber bridging. The dispersion of cotton fibers in slurry is poor, and the fracture toughness decreases with the increase of fibers content. The dispersion of cotton fibers in geopolymer matrix has a great influence on the workability of the new mixture. The addition of 0.7 and 1.0 wt% cotton fibers decreased the workability of the substrate. Increasing water content to overcome this problem may lead to other adverse effects, such as increased porosity and microfractures. This will result in less binding at the fiber–matrix interface, thereby reducing the stress transferred from the matrix to the fibers.

When the first cracking load is greater than the peak load, the strength increases with the deflection, which is the deflection hardening behavior. The PFRGs with flexural hardening have higher bearing capacity after the first cracking. One of the advantages of deflection hardening PFRGs is that it has a higher energy absorption capacity than deflection softening composites. The influence of different types of fibers and different contents on the stress–strain curves of PFRGs are shown in Figure 12a,b [97,116]. It can be seen in Figure 12a that the flexural strength of PFRGs with flax fibers content of 4.1wt % is the highest among all composites. The flexural strength of the PFRGs increased from 4.5 MPa with pure geopolymer to 23 MPa with 4.1 wt% fibers content. Figure 12b shows the same situation. The lower weight of linen fabric makes the multilayer fabric in the composite resistant to shear failure. This results in greater stress transfer between the substrate and the flax fibers, resulting in increased flexural strength. Alzeer et al. [116] used 4–10 vol.% phormium tenax unidirectional reinforced geopolymer. The mechanical properties of the PFRGs increased with the increase of fibers content. When the fibers content is 10 vol.%, the ultimate flexural strength is approximately 70 MPa. This shows that the flexural strength of the pure geopolymer (approximately 5.8 MPa) is significantly increased, and the failure of the PFRG is ductile failure rather than brittle failure.

## 5. Pretreatment of Fibers

Because of its low cost, low density, and good mechanical properties, PFs have attracted more and more attention in reinforced composites. In order to make the fiber have stronger adhesion and better compatibility with the slurry, it is necessary to treat the fiber and modify the surface of the fiber. Alkalization is one of the chemical modification technologies of biological-based materials. The treated PFs have fewer impurities and increase the bonding of their contact surfaces. Figure 13 shows scanning electron microscopy images before and after modification of coir fibers.

As can be observed in Figure 13, the wax, oil, and different impurities in the coir fibers were removed after pretreatment of fibers with 5% NaOH for 24 h. The surface texture of coir fibers were also improved by pretreatment.

Fiber pretreatment can be used in many ways, and different treatment methods have different effects on the properties of PFRGs. Zhou et al. [95] carried out alkali treatment, PVA liquid treatment and oil treatment on the cotton stalk powder PFRGs with different contents. Figure 14a–d show the compression and flexural strength curves of PFRGs with cotton stalk powder (CSP) content under different pretreatment conditions, respectively. According to the particle size is divided into three types: CSP1 (0.1–0.25 mm), CSP2 (0.25–0.5 mm), and CSP3 (0.5–1 mm). With the increase of cotton stalk powder content, the compressive and flexural strength of PFRGs increased first and then decreased, reaching the maximum value when the cotton stalk powder content was 0.7%. The mechanical properties of PFRGs after PVA treatment decreased with the increase of cotton stalk powder content. The powder clusters reduced the compactness of the cementitious matrix and thus the strength. Among all types of PFRGs, the strength of cotton stalk powder geopolymer treated with PVA was the highest, which were 29.2 MPa and 3.3 MPa, respectively, which were 16.4% and 2.6% higher than those untreated. PVA solution can greatly improve the binding properties of powder and matrix. When the amount of cotton stalk powder is fixed, the compressive strength and flexural strength of CSP2 and CSP3 are the highest in the powder treated with alkali. Alkali treatment removes sugar and increases the surface roughness of the powder, helping to fill pores and improve compactness. It was found that alkali treatment was effective, and the compressive strength and flexural strength were increased by 4.8% and 11.5%, respectively.

Similarly, Yan et al. [117] found that after alkali treatment, the compressive strength, flexural strength and toughness of coir fiber reinforced composites increased by 7.1%, 21.4% and 449%, respectively. This is because alkali treatment can promote the precipitation of soluble sugar in advance and affect the polymerization reaction. It was found that the flexural strength of the geopolymer reinforced by rice straw could be significantly improved by both untreated and alkali treated rice straw. However, alkali treatment of rice straw had higher strengthening effect. After curing for 28 days, the flexural strength of rice straw reinforced geopolymer composite with 10% fibers content reached 13.6 MPa [118]. The surface properties of the waste abaca fiber were improved by different chemical treatments to improve its adhesion to the geopolymer matrix. The following factors were considered: (1) pretreatment with NaOH; (2) Immersion time of aluminum salt solution; (3) Final pH value of slurry. The results showed that the fiber pretreated without alkali, soaked in Al_2_(SO_4_)_3_ solution for 12 h and adjusted pH 6 had the highest tensile strength. The results showed that the chemical treatment removed lignin, pectin and hemicellulose, roughened the surface and deposited aluminum compounds. This improves the interfacial binding between the geopolymer matrix and the abaca fibers, while the geopolymer protects the treated fibers from thermal degradation [119]. Figure 15 shows the SEM images of fiber combined with matrix before and after treatment in which Figure 15a,b are waste abaca fibers, and Figure 15c,d are rice straw fibers.

In Figure 15a, for the untreated abaca PFRG, a significant gap between the geopolymer matrix and the fibers was observed, indicating poor interfacial adhesion. This may be due to incompatibility between fibers and the matrix. Circular particles were also observed on the surface, which may be geopolymer precursors or nuclei adhering to the surface of the abaca fibers. On the other hand, in Figure 15b, for the treated abaca PFRG, a narrower gap between the matrix and treated fibers was observed. Zeolite-like particles on the surface of the fibers indicate that reactions have taken place on the fiber surface to form these structures. Abaca fiber aggregation was also observed in other regions of the fracture plane. Although geopolymer and zeolite deposition were observed, which indicates an interaction between the geopolymer matrix and the fiber surface, pull-out sites were also observed, indicating insufficient adhesion of the interface formed in some regions. However, fiber pull-out during loading absorbs energy, thereby improving the flexural strength of the composite [119]. It can be seen from Figure 15c,d that there is obvious fiber peeling surface on the geopolymer matrix, there is a large gap between the fiber and the geopolymer matrix, and the fiber and the geopolymer matrix show signs of debonding. In addition, there are holes after the fibers are pulled out, and local fibers are interwoven and aggregated, as shown in Figure 15c, indicating that the fibers are separated from the matrix to a certain extent and do not absorb the load transmitted by the matrix. Therefore, the strength of untreated straw fiber is low as the treated straw fiber geopolymer [118].

Fonseca et al. [90] analyzed the feasibility of pine fiber, earth palm fiber, razorgrass fiber, and jute fiber as mortar reinforcement materials. The fibers were reinforced by hot water treatment, keratinization, 8% NaOH solution, and hybridization, respectively. The results showed that the hybridization and keratinization promoted the best compatibility between these fibers and the matrix. The alkaline hybridization treatment increased the ductility and mechanical properties of the palm fiber samples because it reorganized the cellulose chains and caused the fibers to split into more fibrils, increasing the direct tensile strength of the fibers from 67.2 MPa to 318.81 MPa.

In addition, fiber properties can be improved by fiber self-handling behavior in geopolymer. The self-treatment process is the addition of PFs to the geopolymer mixture without any pre-alkaline treatment. The self-treatment process is controlled by the alkaline environment in geopolymer system and is also used as the alkaline treatment condition for natural fibers. In the self-treatment method, the pre-alkalization process can be skipped and its mechanical properties are similar to those of the ordinary process. This could replace some or all of the conventional alkaline treatment processes and provide an opportunity for any other natural fiber geopolymer composite to be readily available for mass utilization and practical application [120].

## 6. Limitations and Recommendations

However, PFRGs have some limitations. Compared with traditional fiber, PFs have lower mechanical properties. In alkaline environment, PFs will precipitate some sugars, such as cellulose, hemicellulose, and lignin, which will inhibit the strength development of cementitious materials. The water absorption and variability of PFs are large, and the adhesion of cementitious materials is poor. PFs is easy to be degraded in alkaline matrix, which affects the long-term properties of composites. Hemicellulose in plants is bound to the surface of cellulose, which is easy to dissolve and hydrolyze in alkali solution to produce carbohydrates, which will hinder the polymer solidification in the geopolymer polymerization reaction, and eventually have a negative impact on the mechanical properties of PFRGs.

It is suggested to improve the surface repair and modification of PFs, especially to make full use of the self-treatment properties of PFs. In order to give full play to the green and environmental advantages of PFs reinforcement and geopolymer matrix. At the same time, the interfacial bonding and toughening mechanism of PFs in geopolymer matrix should be further studied. To give full play to the performance advantages of PFs, mix with other types of fiber with complementary properties as a geopolymer reinforcement.

## 7. Conclusions

On the one hand, PFs are energy saving and low cost. On the other hand, they are biodegradable, renewable, and light. From the point of view of sustainable development, PFs can replace synthetic fibers as the reinforcement of geopolymers. However, PFs are less durable and degrade over time. Future studies should consider the long-term durability of PFRGs to evaluate their potential applications in exterior components of construction works.

PFs have good compatibility with geopolymers, and the compatibility of wood fibers is better than that of non-wood fibers;PFs are more effective in improving flexural strength than compressive strength. Adding PFs to geopolymers have the function of toughening and strengthening composites. The higher the content of cellulose, the greater the toughening and strengthening function;At present, physical or chemical methods are commonly used to modify and pretreat PFs to remove sugar in advance. At the same time, it can greatly improve the surface structure of PFs and enhance the bonding ability with the matrix of geopolymers;After alkali treatment, the compressive strength, flexural strength and toughness of some PFRGs increased by 7.1%, 21.4% and 449%, respectively.

## Figures and Tables

**Figure 1 polymers-14-04134-f001:**
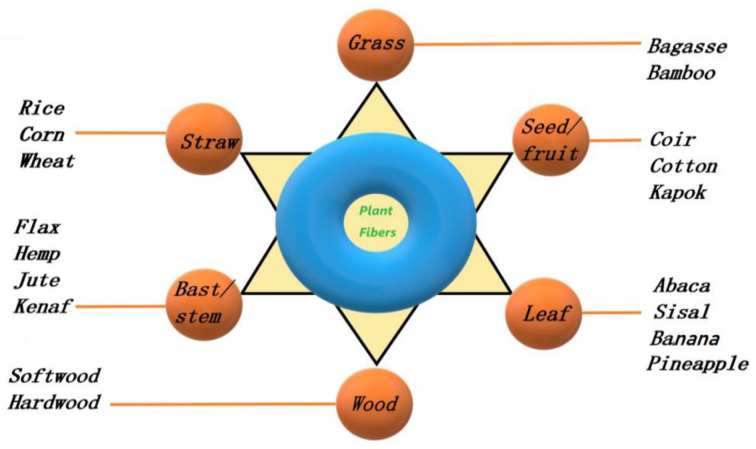
Classification of commonly used PFs.

**Figure 2 polymers-14-04134-f002:**
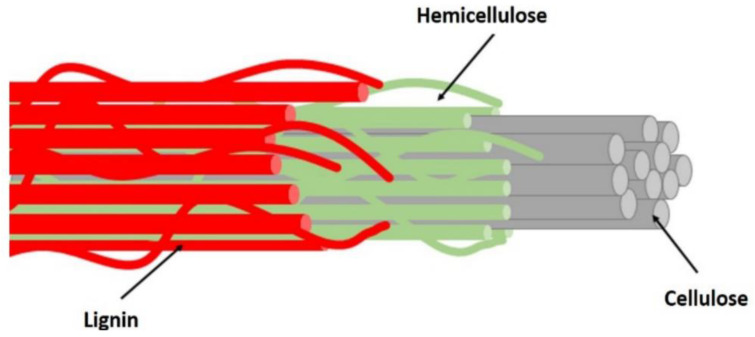
Basic structure of PF [42].

**Figure 3 polymers-14-04134-f003:**
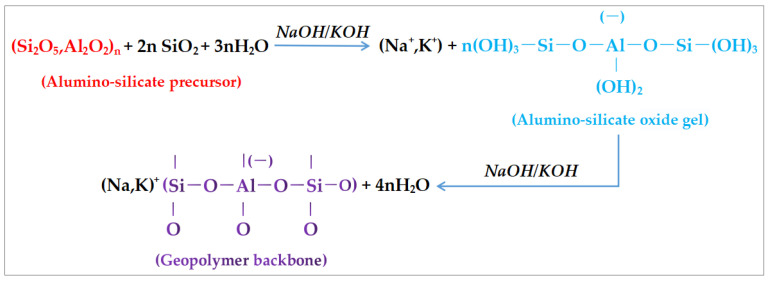
Expression of reaction mechanism of geopolymer.

**Figure 4 polymers-14-04134-f004:**
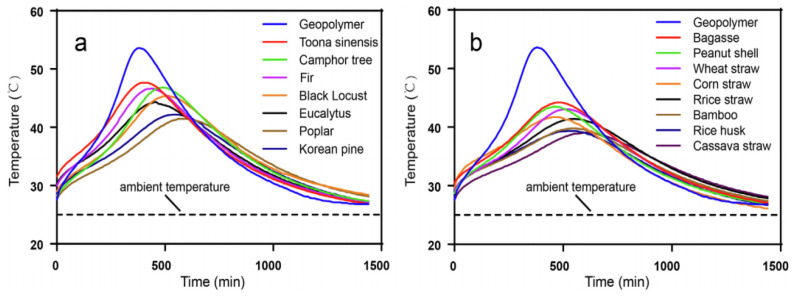
The geopolymerization temperature of PFRG as a function of time. (**a**) Wood fiber; (**b**) Non-wood fiber [78].

**Figure 5 polymers-14-04134-f005:**
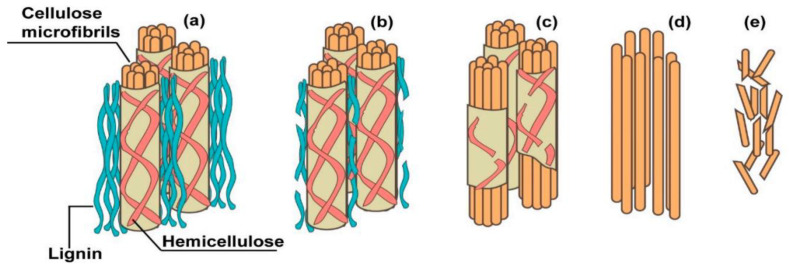
Schematic diagram of alkaline degradation mechanism of PFs. (**a**) major components of PFs; (**b**) Degradation of lignin and part of hemicellulose; (**c**) Degradation of hemicellulose; (**d**) Stripping of cellulose microfibrils; (**e**) Failure of cellulose microfibrils [81].

**Figure 7 polymers-14-04134-f007:**
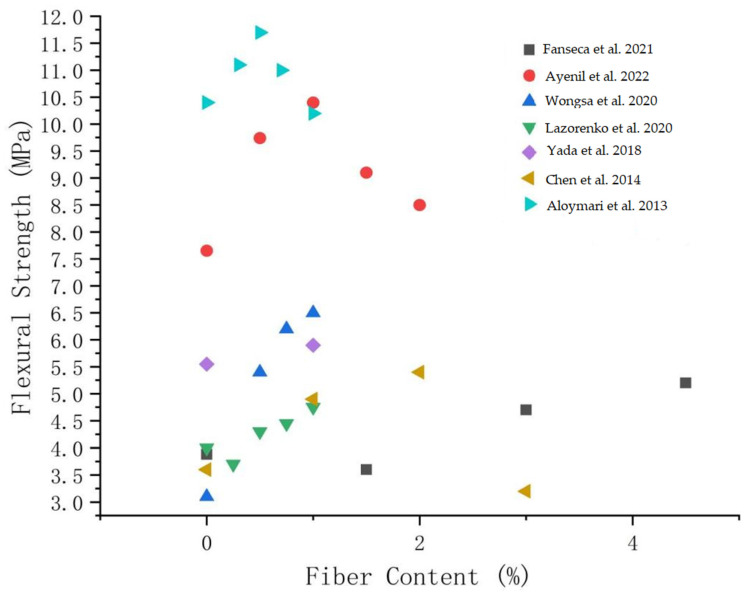
Relationship between fibers content and flexural strength [90,91,92,94,98,101,108].

**Figure 8 polymers-14-04134-f008:**
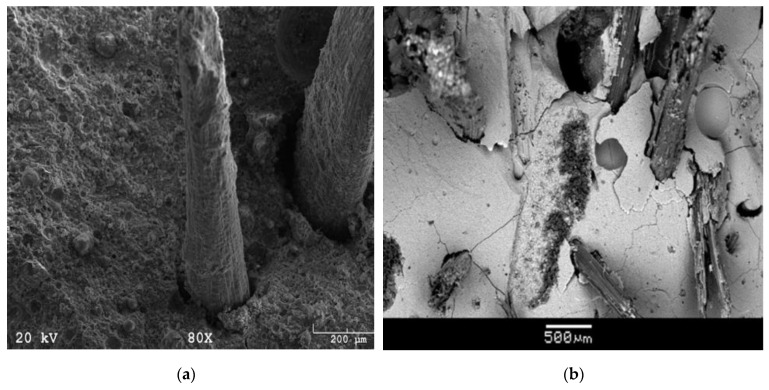
Microstructure of fiber in matrix after bending (**a**) Coir fiber; [98] (**b**) Bamboo fiber [110].

**Figure 9 polymers-14-04134-f009:**
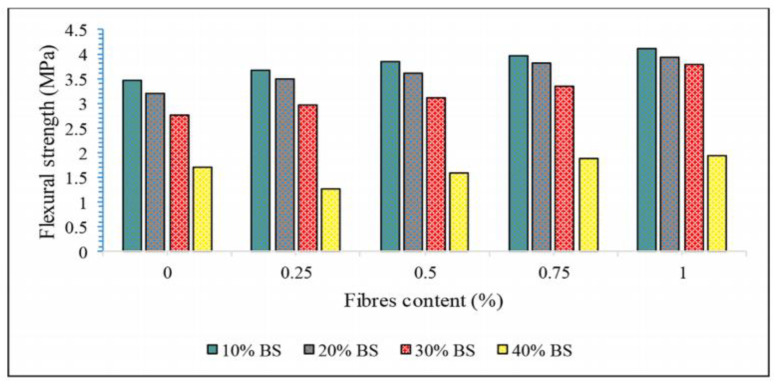
Flexural strength of PFRG with different contents of sisal fiber and bamboo stick(BS) [111].

**Figure 10 polymers-14-04134-f010:**
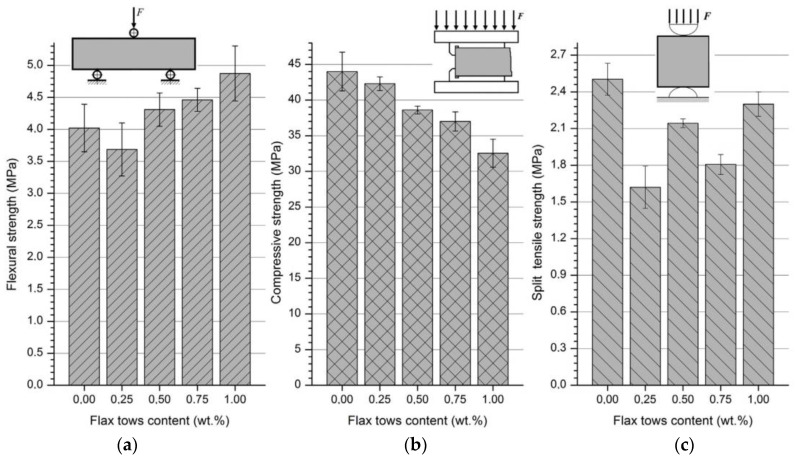
Effect of flax tows content on (**a**) flexural strength, (**b**) compressive strength, (**c**) split tensile strength of PFRGs [94].

**Figure 11 polymers-14-04134-f011:**
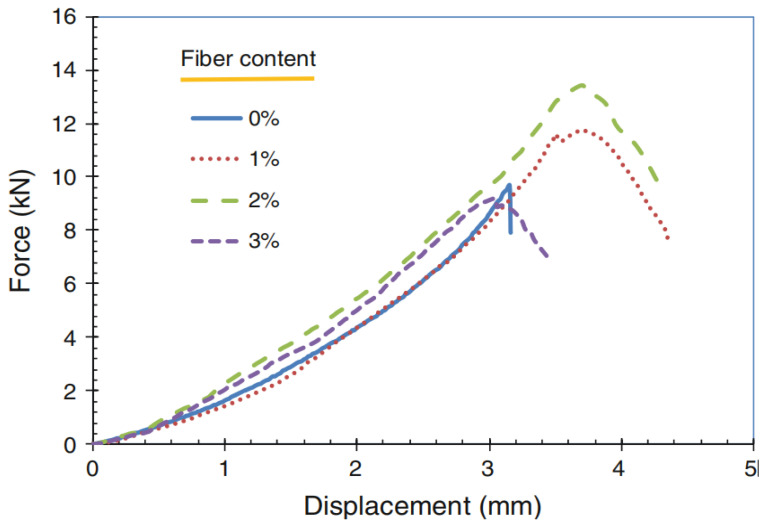
Load and displacement curves of sweet sorghum PFRGs in splitting tensile test [101].

**Figure 12 polymers-14-04134-f012:**
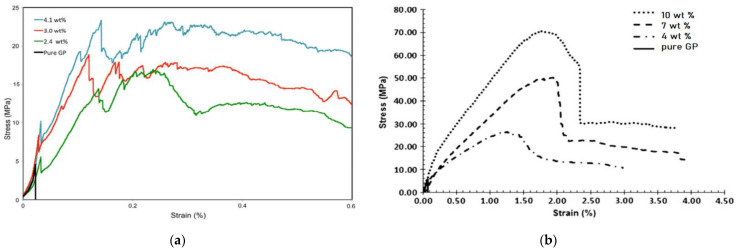
Typical stress–strain curves of pure geopolymer and PFRGs with various fiber contents (**a**) Flax fibers: 0–4.1 wt% [97], (**b**) phormium tenax fibers: 0–10 wt% [116].

**Figure 13 polymers-14-04134-f013:**
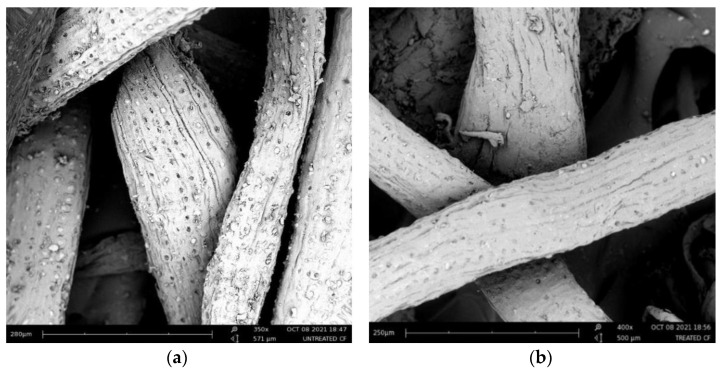
SEM of (**a**) untreated coir fibers and (**b**) pre-treated coir fibers [91].

**Figure 14 polymers-14-04134-f014:**
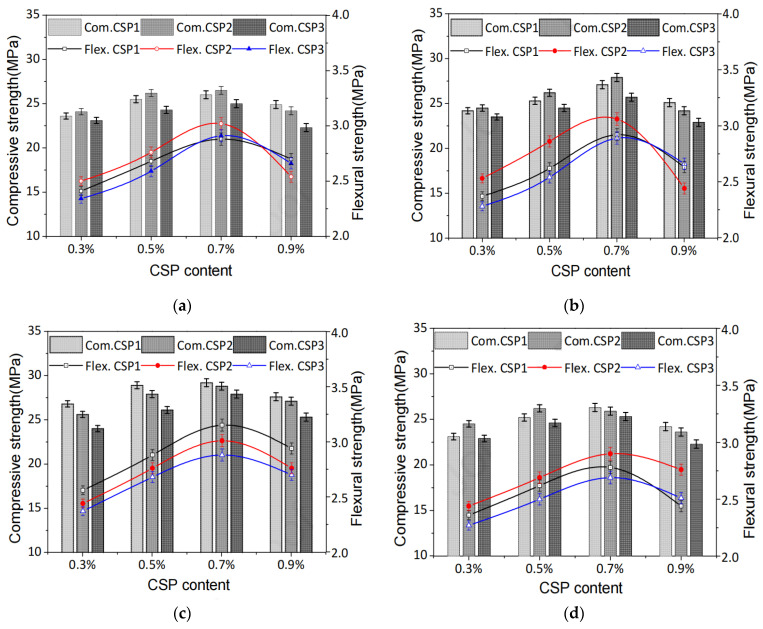
Compression and bending properties of PFRGs with different cotton stalk powder (CSP) content and particle size for 28 days. (**a**) Fiber untreated, (**b**) alkali treated, (**c**) PVA liquid treated and oil treated (**d**) [95].

**Figure 15 polymers-14-04134-f015:**
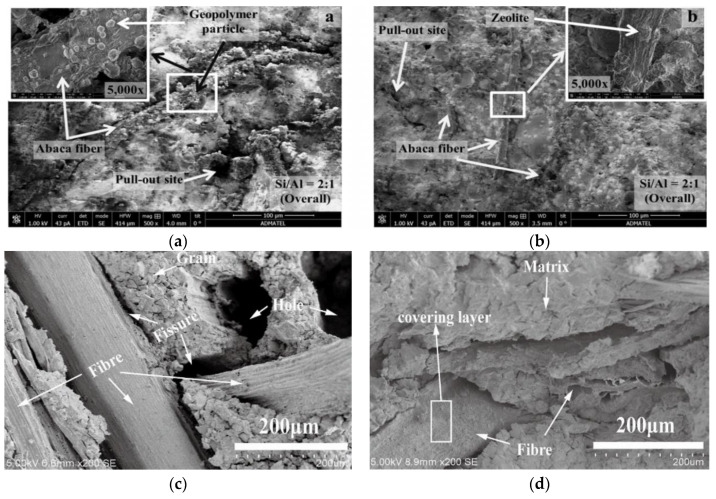
SEM images of composite fractured surfaces: (**a**) reinforced with untreated abaca, (**b**) Treated abaca [119], (**c**) Untreated straw, (**d**) Treated straw [118].

**Table 1 polymers-14-04134-t001:** Mechanical properties of typical PFs [48,49,50,51,52,53,54,55,56,57,58,59,60,61,62,63,64,65,66,67,68,69,70].

Fiber Type	Fiber Name	Density /g·cm^−^^3^	Tensile Strength/MPa	Tensile Modulus/GPa	Elongation at Break/%	Ref.
Bast	Flax	1.50	800–1500	27.60–80.00	1.2–3.2	[48,49,50,51,52,53,54]
	Hemp	1.48	550–900	70.00	2.0–4.0	[49,50,51,52,53,55]
	Jute	1.46	393–800	10.00–30.00	1.5–1.8	[54]
	Kenaf	1.45	930	53.00	1. 6	[55]
	Ramie	1.50	220–938	44.00–128.00	2.0–3.8	[54,58]
Leaf	Abaca	1.50	400	12.00	3.0–10.0	[59]
	Sisal	1.45	530–640	9.40–22.00	3.0–7.0	[53,58,60,61,62,63]
	Banana	1.35	600	17.85	3.4	[58]
	Pineapple	0.80-1.60	400–627	1.40	14.5	[54,64]
	Coconut	1.15	500	2. 50	20.0	[65]
Seed/Fruit	cotton	1.60	287–597	5.50–12.60	7.0–8.0	[65,67,68,69,70]
	Coir	1.20–1.35	120–200	19.00–25.00	10.0–25.0	[51,53,64]
Grass	bamboo	1.10	500	35.91	1.4	[51,65]
Wood	Soft wood	1.50	1000	40.00	4.4	[65]

## Data Availability

Not applicable.

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
