# Peer review of "The Mechanical Properties of Plant Fiber-Reinforced Geopolymers: A Review"

_polymers, 2022, doi:10.3390/polym14194134_

Round 1
Reviewer 1 Report
Manuscript ID: polymers-1933333
Title: Mechanical properties of plant fiber-reinforced geopolymers: A review
Journal: Polymers
Comments to authors:
General: In this paper, the recent studies on mechanical properties of plant fiber-reinforced geopolymers (PFRGs) are systematically reviewed. The characteristics of plant fibers and the composition, structure and properties of geopolymers are reviewed. The compatibility of geopolymer material and plant fiber and the degradation of fiber in the substrate are analyzed. The effects of plant fibers on the compressive, tensile and bending properties of geopolymers are summarized. Meanwhile, the modification of plant fiber is also discussed. Through the comprehensive analysis of PFRGs, the existing problems and future development direction of PFRGs at present are pointed out.
Recommendation: This manuscript can be accepted after a Minor Revision. The authors are requested to address the following comments for the improvement of the manuscript:
1) The Abstract should be enriched with the brief details of the methodology.
2) Background should be briefly presented (not explained) in the Abstract.
3) Use only one tense of English in the Abstract.
4) English proofreading should be done for grammar and typos.
5) The novelty, scope, and significance of the present work should be highlighted in the last paragraph of the Introduction section.
6) The authors are recommended to add latest literature review on such composites and relevant works to update the literature review section. Following papers are very helpful in this regards:
https://doi.org/10.1007/s41779-022-00764-9
https://doi.org/10.3390/polym13213852
7) What is the need for this work?
8) Is this work helpful for practical applications? Which applications?
9) The literature review should be improved by adding latest references and discussion.
10) Discussion of microstructure should be more elaborated using Scanning Electron Microscopy (SEM) of composites.
11) The authors should discuss more mechanical properties of the examined composites.
12) Conclusions should be refined and briefly presented. Some numerical results should be added.
13) Limitations of the studied composites should be added.
14) The authors should add the future recommendations based on the present study.
Author Response
Reviewer 1:
General: In this paper, the recent studies on mechanical properties of plant fiber-reinforced geopolymers (PFRGs) are systematically reviewed. The characteristics of plant fibers and the composition, structure and properties of geopolymers are reviewed. The compatibility of geopolymer material and plant fiber and the degradation of fiber in the substrate are analyzed. The effects of plant fibers on the compressive, tensile and bending properties of geopolymers are summarized. Meanwhile, the modification of plant fiber is also discussed. Through the comprehensive analysis of PFRGs, the existing problems and future development direction of PFRGs at present are pointed out.
Recommendation: This manuscript can be accepted after a Minor Revision. The authors are requested to address the following comments for the improvement of the manuscript:
1) The Abstract should be enriched with the brief details of the methodology.
Thank you very much for reviewing our manuscript in your busy schedule. Based on your valuable suggestions, we have carefully revised our manuscript. (The revised page 1, lines 27-29)
2) Background should be briefly presented (not explained) in the Abstract.
Thank you very much for reviewing our manuscript in your busy schedule. Based on your valuable suggestions, we have carefully revised our manuscript. (The revised page 1, lines 21-22)
3) Use only one tense of English in the Abstract.
Thank you very much for reviewing our manuscript in your busy schedule. Based on your valuable suggestions, we have carefully revised our manuscript. (The revised page 1, lines 23, 28)
4) English proofreading should be done for grammar and typos.
Thank you very much for reviewing our manuscript in your busy schedule. Based on your valuable suggestions, we have carefully revised our manuscript. English proofreading for grammar and typos has been carefully revised. The goal point of this manuscript is achieved successfully.
5) The novelty, scope, and significance of the present work should be highlighted in the last paragraph of the Introduction section.
Thank you very much for reviewing our manuscript in your busy schedule. Based on your valuable suggestions, we have carefully revised our manuscript. (The revised page 3, lines 117-122)
6) The authors are recommended to add latest literature review on such composites and relevant works to update the literature review section. Following papers are very helpful in this regards: A scientometric review on mechanical and durability performance of geopolymer Paste: Effect of various raw materials; Enhancement of mechanical and toughness properties of carbon fiber-reinforced geopolymer pastes comprising nano calcium oxide; Mechanical Performance of Geopolymer Composites Containing Nano-Silica and Micro Carbon Fibers; Experimental study of the mechanical properties and microstructure of geopolymer paste containing nano-silica from agricultural waste and crystalline admixtures; Mechanical, Fracture and Microstructural Assessment of Carbon Fiber-Reinforced Geopolymer Composites Containing Na2O; Experimental Investigation on the Mechanical and Fracture Evaluation of Carbon Fiber-Reinforced Cementitious Composites with Nano-Calcium Carbonate.
Thank you very much for your review. According to your suggestion, we have carefully add the latest references. (The revised page 19).
7) What is the need for this work?
Geopolymers can be effectively used as a practical alternative to cement to avoid a high carbon footprint and develop sustainable concrete construction. Plant fiber is rich in source, has good properties, and has good compatibility with geopolymers. This work could lead to practical applications in environmental sustainability.
8) Is this work helpful for practical applications? Which applications?
This work is a comprehensive summary of plant fiber-reinforced geopolymer composites, which can lay a foundation for further study of its working mechanism and engineering application.
9) The literature review should be improved by adding latest references and discussion.
Thank you very much for reviewing our manuscript in your busy schedule. Based on your valuable suggestions, we have carefully revised the latest references and discussion. (The revised page 19)
10) Discussion of microstructure should be more elaborated using Scanning Electron Microscopy (SEM) of composites.
Thank you very much for reviewing our manuscript in your busy schedule. Based on your valuable suggestions, we have carefully revised our manuscript. (The revised page 16, lines 570-581)
11) The authors should discuss more mechanical properties of the examined composites.
Thank you very much for reviewing our manuscript in your busy schedule. Based on your valuable suggestions, we have carefully revised our manuscript. (The revised page 12, lines 431-447; The revised page 14, lines 5025-544)
12) Conclusions should be refined and briefly presented. Some numerical results should be added.
Thank you very much for reviewing our manuscript in your busy schedule. Based on your valuable suggestions, we have carefully revised our manuscript. (The revised page 18, lines 634-645)
13) Limitations of the studied composites should be added.
Thank you very much for reviewing our manuscript in your busy schedule. Based on your valuable suggestions, we have carefully revised our manuscript. (The revised page 17, lines 609-619)
14) The authors should add the future recommendations based on the present study.
Thank you very much for reviewing our manuscript in your busy schedule. Based on your valuable suggestions, we have carefully revised our manuscript. (The revised page 18, lines 620-626)
In addition, we have also revised other parts of the article according to your review suggestions (highlighted parts in the manuscript).
According to your suggestion, we have carefully and comprehensively revised the manuscript.
Finally, thank you again for your wonderful review of our article in your busy schedule.
Reviewer 2 Report
Dear Author,
Please find my comments for improvement, do perform the corrections and send back the article for further processing.
A general search with the title fetched very less results and this article has novelty.
Authors did a good work on preparing the manuscript with primary focus on geoplymers, please add one paragraph to indicate what motivated you to write this particular topic based on science.
Can Natural fiber a good word instead of plant fibers? Plant fiber is somewhat awkward, Natural fiber reinforced composites are already existing and a well known word.
Line 22: Is this a systematic review? Or General review?
Line 26: Modifications done on fibres were also discussed.
Line 34-37 is not relevant to this context.
Upto line 60 most of the readers know about geopolymers, focus on NFR Geopolymer please.
Fig 1: Why banana fiber is missed? There are works related to this also, please check.
Line 123: Different fibers means what?
Line 120: Is biodegradability a good property for being considered as a reinforcing material?
My concern is Geopolymer generates heat energy while preparation and curing, will this not affect fibers is in my mind, please look into it.
Fig 6 please correlate it with the thickness and length of fibre used since it influences the strength for sure.
Line 302, year is missing in reference
Fig 11: is it common for all fibers? Percentage of addition.
Fig 12: Interpretation issue will arise increase logic in the figure.
Before conclusion I suggest authors interpretation and recommendations in one or two paragraphs
Mixing of fibers is not dealt in this paper, this is one important area, mixing fibers with geoplymer is not easy, please check and update if possible.
Author Response
Reviewer 2:
Please find my comments for improvement, do perform the corrections and send back the article for further processing.
A general search with the title fetched very less results and this article has novelty.
Authors did a good work on preparing the manuscript with primary focus on geoplymers, please add one paragraph to indicate what motivated you to write this particular topic based on science.
Thank you very much for reviewing our manuscript in your busy schedule. Based on your valuable suggestions, we have carefully revised our manuscript.
Can Natural fiber a good word instead of plant fibers? Plant fiber is somewhat awkward, Natural fiber reinforced composites are already existing and a well known word.
Thank you very much for reviewing our manuscript in your busy schedule. Based on your valuable suggestions, we have carefully checked our manuscript. Natural fiber is a really good word, and it's a well known word. Plant fiber is the main component of natural fibers. This manuscript only discusses the part of plant fiber in natural fibers, not animal fiber and natural mineral fiber in natural fibers. Many thanks for the reviewer's correction. In the next step, we will further discuss the common issues of natural fibers as suggested by reviewers.
Line 22: Is this a systematic review? Or General review?
Thank you very much for your review. According to your suggestion, we have carefully rewritten the sentence. (The revised page 1, lines 21-23).
Line 26: Modifications done on fibres were also discussed.
Thank you very much for your review. According to your suggestion, we have carefully rewritten the sentence. (The revised page 1, lines 28-29).
Line 34-37 is not relevant to this context.
Thank you very much for your review. According to your suggestion, we have carefully rewritten the sentence. (The revised page 1, lines 35-37).
Upto line 60 most of the readers know about geopolymers, focus on NFR Geopolymer please.
Thank you very much for your review. According to your suggestion, we have carefully rewritten the sentence. (The revised page 2, lines 66-68).
Fig 1: Why banana fiber is missed? There are works related to this also, please check.
Thank you very much for your review. According to your suggestion, we have carefully rewritten the sentence. (The revised page 3, lines 137-138).
Line 123: Different fibers means what?
Thank you very much for your review. According to your suggestion, we have carefully rewritten the sentence. (The revised page 3, lines 132-133).
Line 120: Is biodegradability a good property for being considered as a reinforcing material?
Thank you very much for your review. Biodegradability is not a good property as a reinforcement material. The purpose of our analysis of fiber degradation mechanism is to prevent fiber biodegradation in the matrix.
According to your suggestion, we have carefully rewritten the sentence. (The revised page 3, lines 124-125).
My concern is Geopolymer generates heat energy while preparation and curing, will this not affect fibers is in my mind, please look into it.
Thank you very much for your review. According to your suggestion, we have carefully added this content. (The revised page 7, lines 242-247).
Fig 6 please correlate it with the thickness and length of fibre used since it influences the strength for sure.
Thank you very much for your review. According to your suggestion, we have carefully rewritten the content. (The revised page 8, lines 305-307).
Line 302, year is missing in reference
Thank you very much for your review. According to your suggestion, we have carefully rewritten the content. (in Reference).
Fig 11: is it common for all fibers? Percentage of addition.
Thank you very much for your review. According to your suggestion, we have carefully rewritten the content. (The revised page 13, line 475).
Fig 12: Interpretation issue will arise increase logic in the figure.
Thank you very much for your review. According to your suggestion, we have carefully rewritten the content. (The revised page 14, lines 494-497; The revised page 14, line 510).
Before conclusion I suggest authors interpretation and recommendations in one or two paragraphs
Thank you very much for your review. According to your suggestion, we have carefully rewritten the content. (The revised page 17, lines 609-626).
Mixing of fibers is not dealt in this paper, this is one important area, mixing fibers with geoplymer is not easy, please check and update if possible.
Thank you very much for your review. According to your suggestion, we have carefully rewritten the content. (The revised page 7, lines 242-247).
In addition, we have also revised other parts of the article according to your review suggestions (highlighted parts in the manuscript).
According to your suggestion, we have carefully and comprehensively revised the manuscript.
Finally, thank you again for your wonderful review of our article in your busy schedule.
Reviewer 3 Report
The paper presents an interesting approach based on the Mechanical properties of plant fiber-reinforced geopolymers:A review. However, the innovation of the current research work should be further highlighted and emphasized. At the same time, the authors should consider the following comments to greatly improve the quality of the paper.
1. Kindly unify the English style used in the article. Currently, there is a mix between UK English and US English in several instances, such as the use of "Fibres" and "Fibers".
2. The introduction needs to be improved by relating to the mechanics of the studied materials and their mechanical characteristics. The references to be included are: 10.1177/0021998318790093, 10.1016/j.polymertesting.2017.09.009, 10.1016/j.compstruct.2021.114698, 10.1177/0731684417727143, 10.1002/app.46770, 10.1016/j.porgcoat.2022.107015.
3. In the compressive strength and flexural strength figures that show the performance of relevant materials which were done by other researchers, require to be cited in the caption text of these figures.
4. The tensile strength section is very short. It has to be extended.
5. Generally, in each of the mechanical properties investigated, there should a unified format that has to be followed for each sub-section such as: Morphology, Property trends and analysis/critcisim that highlights the gaps, similarities or dissimilarities between studies.
6. It is observed that in several sub-sections, there are few studies that take the attention of the authors. This is a very broad subject and in a review paper, there must be a wide and comprehensive overview on much more groups of articles. This is a very critical missing element in this review. Also, the absence of extended review tables lowers the quality of this review.
7. The conclusion (should be re-numbered as it is currently in correct!) needs to be modified to summarize the research outcomes in short statements with clear observations.
Author Response
Reviewer 3:
The paper presents an interesting approach based on the Mechanical properties of plant fiber-reinforced geopolymers:A review. However, the innovation of the current research work should be further highlighted and emphasized. At the same time, the authors should consider the following comments to greatly improve the quality of the paper.
- Kindly unify the English style used in the article. Currently, there is a mix between UK English and US English in several instances, such as the use of "Fibres" and "Fibers".
Thank you very much for your review. According to your suggestion, we have carefully rewritten the content. (The revised page 6, line 226).
- The introduction needs to be improved by relating to the mechanics of the studied materials and their mechanical characteristics. The references to be included are: 10.1177/0021998318790093, 10.1016/j.polymertesting.2017.09.009, 10.1016/j.compstruct.2021.114698, 10.1177/0731684417727143, 10.1002/app.46770, 10.1016/j.porgcoat.2022.107015.
https://doi.org/10.1016/j.porgcoat.2022.107015
Thank you very much for your review. According to your suggestion, we have carefully rewritten the references.
- In the compressive strength and flexural strength figures that show the performance of relevant materials which were done by other researchers, require to be cited in the caption text of these figures.
Thank you very much for your review. According to your suggestion, we have carefully rewritten the content. (The revised page 8, lines 308-309; The revised page 10, line 384).
- The tensile strength section is very short. It has to be extended.
Thank you very much for your review. According to your suggestion, we have carefully rewritten the content. (The revised page 12, lines 430-431; The revised page 12, lines 445-447).
- Generally, in each of the mechanical properties investigated, there should a unified format that has to be followed for each sub-section such as: Morphology, Property trends and analysis/critcisim that highlights the gaps, similarities or dissimilarities between studies.
Thank you very much for your review. According to your suggestion, we have carefully rewritten the content. (The revised page 7, lines 235-510).
- It is observed that in several sub-sections, there are few studies that take the attention of the authors. This is a very broad subject and in a review paper, there must be a wide and comprehensive overview on much more groups of articles. This is a very critical missing element in this review. Also, the absence of extended review tables lowers the quality of this review.
Thank you very much for your review. According to your suggestion, we have carefully rewritten the content. We have increased the number of references, modified and added relevant content.
- The conclusion (should be re-numbered as it is currently in correct!) needs to be modified to summarize the research outcomes in short statements with clear observations.
Thank you very much for your review. According to your suggestion, we have carefully rewritten the content. (The revised page 18, lines 634-645).
In addition, we have also revised other parts of the article according to your review suggestions (highlighted parts in the manuscript).
According to your suggestion, we have carefully and comprehensively revised the manuscript.
Finally, thank you again for your wonderful review of our article in your busy schedule.
Round 2
Reviewer 3 Report
The article can be accepted after the authors add these references:
10.1016/j.polymertesting.2017.09.009,
10.3390/polym14132662,
10.1016/j.jiec.2022.06.023.